# Otolith Weight as an Estimator of the Age of *Seriola lalandi* Valenciennes, 1833 (Carangidae), in the Southeastern Pacific

**DOI:** 10.3390/ani12131640

**Published:** 2022-06-26

**Authors:** Tchimanda Simeão Imbo Ndjamba, Miguel Araya, Marcelo Enrique Oliva

**Affiliations:** 1Magíster Ecología Sistemas Acuáticos Programme, Universidad de Antofagasta, Antofagasta 3580000, Chile; tchimandasimeao@hotmail.com; 2Faculdade de Ciências das Pescas, Universidade do Namibe, Moçâmedes 12004, Angola; 3Facultad de Recursos Naturales Renovables, Universidad Arturo Prat, Iquique 1110939, Chile; maraya@unap.cl; 4Instituto de Ciencias Naturales Alexander von Humboldt, Universidad de Antofagasta, Antofagasta 1270300, Chile; 5Millenium Institute of Oceanography, Universidad de Concepción, Concepción 78349, Chile

**Keywords:** age and growth, otolith, yellowtail kingfish, seasonalized growth function

## Abstract

**Simple Summary:**

The healthy management of fishery resources requires the evaluation of a series of population attributes, such as mortality, fecundity, spawning biomass, recruitment and the age at which fish reach sexual maturity. All these attributes depend on the age of fish. Then, the adequate administration and management of these resources require estimators of fish age. Age is estimated by ring counts in the otoliths, which are hard and calcified structures responsible for the balance of fish; however, this is an expensive and time-consuming methodology. An alternative is the use of otolith weight due to its expected relationship with fish age. Yellowtail king fish is a valuable resource, which arrives at the Chilean northern coast in the summer (southern hemisphere). Many aspects of the biology of this fish, such as age and growth, remain unknown. In this study, we estimated the age and growth using otolith weight, which was measured from fish with a known age, and we calculated parameters explaining growth using four different models. The best model was the seasonalized von Bertalanffy growth function, which takes into account that fish do not grow at the same rate year round.

**Abstract:**

The age and growth of fish populations is a critical issue for stock assessment, population dynamics and fishery management. Spawner biomass, mortality, growth, recruitment and age at maturity can be misconstrued if the age estimator is inaccurate. Age can be estimated by annuli count, but this requires expertise and is expensive. Otolith weight (OW) is a good indicator of how long a fish has lived, because OW increases during an individual’s life. *Seriola lalandi* is a migratory fish and is an important resource for local fishermen in northern Chile. Aspects of its biology, i.e., age and growth, remain unknown, at least for the population annually arriving in northern Chile. Fish of a known age (*n* = 105; from 5.5 to 25.7 cm in FL) from yellowtail aquaculture facilities at Universidad de Antofagasta allowed us to estimate the growth rate of OW, and fish obtained from local fishermen (*n* = 71; from 37.9 to 109 cm in FL) allowed us to estimate the age and growth of *S. lalandi* from the analysis of OW, without the need for calibration. The following four models were fitted with the known ages and fork lengths: the von Bertalanffy growth function, the Gompertz and logistic models and the seasonalized von Bertalanffy growth function. The latter model showed the best adjustment (according to the Akaike information criterion) with the following parameters: *L*_∞_, 98.58 cm.; *K*, 0.59; *t*_0_, 0.07; *t*_s_, 0.84; and C, 0.97.

## 1. Introduction

An accurate estimation of the age and growth of fish populations is a critical issue in stock assessment, population dynamics and successful fishery management [1,2]; specifically, spawner biomass, mortality, growth, recruitment and age at maturity in particular can be misconstrued if the age estimators are inaccurate [3,4].

Age structures in fish populations have traditionally been estimated by interpreting marks in bone structures such as scales and otoliths; the opercular bones of fish [4,5] have also been used, as has the analysis of annual rings in the otolith, one of the most common methodologies, which requires an initial preparation of the otolith (embedding, sectioning and polishing), and a microscopic examination and counts of the annuli [6]. This process is generally time consuming, and interpreting the annuli requires expertise; consequently, age determination from hard structures is very expensive [7] and is highly subject to the researcher’s skills [8]. Age estimates are typically accepted when readers attain some minimum coefficient of the variation benchmark, although this does not imply that the age is accurate [3]. Age validation methods increase both precision and the accuracy of age estimates [9] but require the use of either the mark-recapture method or the regular sampling of fish over relatively long time periods, which is not always possible [10].

Otoliths are biomineralized concretions of calcium carbonate and other minor elements that are metabolically inert. In teleosts, the inner ears are multi-sensory, stato-acoustic organs with basic vestibular and acoustic functions, and they are critical structures in the perception of angular and linear acceleration. Each inner ear is composed of three semicircular canals, three end organs and three otolith organs (sacculus, utriculus and lagena); three otoliths are located inside: sagitta, lapillus and asteriscus. The otoliths act as transducers of acoustic and vestibular signals to the fish nervous system [11]. The otolith shape (otolith contour) provides us with important information related to paleontological, ictiological and ecological sciences, especially in food web studies [12], and is an efficient tool for fish stock discrimination [13], but otolith chemistry has also indicated large-scale connectivity among fish populations [14]. Otolith growth is continuous over the fish life [15].

The relation between otolith weight (OW) and fish age is based on the growth of the otolith, with a linear relationship between both variables; otolith growth continues but in weight [15]. In addition, this relationship states that larger fish (for a given age class) generally have heavier otoliths than smaller fish of the same age class [3]. Consequently, the OW is a good indicator of how long a fish has lived. The direct relationship between fish otolith weight and age has been studied since 1990 [16,17,18], and this relationship has great potential for estimating the age and age structure in fish [18], simply because OW increases during the whole life of an individual, unlike fish length or otolith size [19]. It is a faster and inexpensive method and has been applied to a variety of fishes from different habitats and regions worldwide [20]. Otolith weight could be a valuable criterion as an age determination technique that is objective, economic and easy to perform, compared to traditional methods that have been defined ‘‘as much an art as a science” [21]. A meta-analysis [16] suggested that otolith weight is a good predictor for the age estimation of fish. Specifically, calibration would require precise and exact estimations of the age of only a relatively small number of fish that cover the whole range of otolith weights and ages present in the samples [22].

The use of OW requires a calibration stage, implying that age estimated based on annuli counting can be considered as the ‘‘true’’ age, and that age calculated using otolith weight can be considered as the ‘‘estimated’’ age [21]. This methodology has been widely used when OW is used as an estimator of age (e.g., among many others in the last decade, [5,6,10,21,23,24]). Calibration can be performed using other methodologies and not only with annuli counts; for instance, the persistence and progression of modes in otolith-weight frequency distributions suggest a direct relationship between otolith weight and fish age [22,25]. A different approach was developed [26] following changes in otolith-weight distribution of pilchard *Sardinops sagax neopilchardus* for a captive population and wild-caught fish, and the modes in the otolith-weight frequency distributions appeared to persist and progress in a fashion that was generally consistent with them representing different year classes. The relation between OW and fish age in lake trout *Salvelinus namaycush* was analyzed using a reference sample reared in a hatchery with a known age of 14–16 months [3]. Consequently, both approaches were independent of a “true age” assignation from annuli counts. 

The relationship between the age and growth of members of the genus *Seriola* have been estimated from different localities [27,28,29,30,31,32,33,34]. The maximum age estimated for *S. lalandi* Valenciennes, 1833, ranged between 7 and 21 years. Otolith ring counts suggested a maximum age of 7 years for males and 8 years for females [34], whereas age estimated based on the analysis of scales varied between 9 and 12 years [28,30] and between 9 and 11 years when the marks in bones (vertebrae) were studied [30,33]. Finally, an approach based on data covering older fish representative of all ages suggested that *S. lalandi* can reach 21 years [32].

*Seriola lalandi* is a highly migratory pelagic fish that is widely distributed in temperate and subtropical waters around the world. Along the south-eastern Pacific Coast, it arrives annually in summer (between 20° S and 30° S) [35] and is an important resource for local fishermen in northern Chile. Many aspects of its biology, i.e., age and growth, remain unknown, at least for the population arriving annually in northern Chile. Universidad de Antofagasta has hatchery facilities for the experimental development of the yellowtail kingfish aquaculture. From the hatchery facilities, we obtained a sample of fish of a known age, which allowed us to estimate the growth rate of OW. Our goal was to estimate the age and growth of *S. lalandi* from the analysis of OW, without the need for calibration using annuli counts or another methodology.

## 2. Materials and Methods

In January–April 2018, we sampled a total of 71 specimens of *S. lalandi* from the fish market at Antofagasta, in northern Chile (23°20′ S). Additionally, we sampled 105 specimens of a known age from the aquaculture facilities at Universidad de Antofagasta. The fork length (FL) for each fish was measured to the nearest centimeter, and otoliths were extracted, cleaned and stored in tagged vials; broken otoliths were discarded. Then, the otoliths were weighed, without being dried in an oven, with an analytic balance (to the nearest 0.0001 gr). To test if the left and right otoliths differed significantly, a “t- test” for paired data was used [36] for 132 pairs of otoliths.

Due to the expected linear relationship between the age and weight of the otolith [20], to estimate age from OW, the following expression was used:(1)$$t_{i}=\frac{OW_{i}}{OW_{mgr}}$$
where *t_i_* is the estimated age (years) of the *i*th fish, *OW_i_* is the weight of the otolith of the *i*th fish (in mg) and *OW_mgr_* is the mean growth rate of the otolith (mg/year). Due to the absence of growth differences between sexes for this species [34,37], age was estimated for the whole sample. *OW_mgr_* was estimated as the slope of the regression between *OW* and the known age of captive fish from aquaculture facilities. 

Four growth models were adjusted with the known age and FL as previously suggested [38,39].

Von Bertalanffy growth function (vBGF):(2)$$L_{t}=L_{\infty }\left(1-e^{-K\left(t-t_{0}\right)}\right)$$
where *L_t_* is the fork length at age *t*, *L*_∞_ is the asymptotic fork length, *K* is the growth coefficient (1/time) and *t*_0_ is the theoretical age when the length is zero (time).

Gompertz growth function (GZ):(3)$$L_{t}=L_{\infty }e^{-e^{-g\left(t-t_{0}\right)}}$$
where *L_t_* and *L*_∞_ are as in the von Bertalanffy growth function, *g* is the instantaneous rate of growth when *t* = *t*_0_ and *t*_0_ is the age at which the absolute rate of increase in length begins to decrease.

Logistic growth function (LG):(4)$$L_{t}=\frac{L_{\infty }}{\left(1+e^{-g(t-t_{0}}\right)}$$
where *L_t_* and *L*_∞_ are as in the vBGF, *g* is the instantaneous rate of growth when the length tends to 0 and *t*_0_ is the age at which the absolute rate of increase in length begins to decrease.

Seasonalized von Bertalanffy growth function (SvBGF):(5)Lt=L∞(1−e−K(t−t0)−CK2π[sin(2π(t−ts))−sin(2π(t0−ts))])
where *L*_∞_, *K* and *t*_0_ are as in the vBGF, C is the amplitude of the growth oscillation and *t_s_* is the time between *t* = 0 and the start of a sinusoid growth oscillation.

The growth models were fitted using the FSA package with R statistical language [40,41].

The best model explaining the growth of *S. lalandi* was selected according to the Akaike information criterion [42].

## 3. Results

Specimens from the aquaculture facilities at Universidad de Antofagasta ranged from 5.5 to 25.7 cm in FL and had known ages from 99 to 329 days (from 0.27 to 0.9 years) (Appendix A).

Specimens caught by local fishermen ranged from 37.9 to 109 cm in FL. Otoliths from 71 specimens where used. For both wild and captive fish, the mean weights of the left and right otoliths did not show significant differences (two-tailed tests = 1.978; *p* = 0.958; df = 131) for the whole sample.

The slope of the regression between OW and FL for captive fish, that is, otolith growth rate, was 3.044 mg/year (*n* = 105; SE = 0.231; r^2^ = 0.629). Using Eq. 1, the estimated age of wild fish ranged from 0.8 to 5.7 years.

The estimated growth parameters obtained from the non-linear fit are given in Table 1. According to the Akaike information criterion, the model that best fit the growth of *S. lalandi* from the south-eastern Pacific was the seasonalized von Bertalanffy growth function (Figure 1).

## 4. Discussion

Age assignments are among the most important biological measures in fishery management [9]; spawner biomass, mortality, growth, recruitment and age at maturity in particular can be misconstrued if age estimators are inaccurate [3,4]. Consequently, biological reference points, which are used in inferring stock status in fisheries and are the targets or thresholds in fishery management, can be affected by an accurate estimate of age and, particularly, individual growth rates [39].

As previously stated [40], the development of the standard von Bertalanffy growth function (vBGF) and the seasonalized SvBGF, as well as the logistic and Gompertz models, makes it possible to estimate population parameters such as asymptotic length (*L_∞_*) and growth coefficient (*K*), under the condition that size at the age is known. 

Age structures in fish populations have traditionally been estimated by interpreting marks in hard structures, such as scales and otoliths, as well as the opercular bones of fish [4,5]. The age structure is often derived by interpreting annual rings in sectioned otoliths from several individuals, giving a relative proportion of the number of specimens in each age class; however, two main disadvantages of this methodology are evident: a consistent age reading requires a certain degree of skill, which mainly depends on the reader’s experience, and the method is time-consuming and thus expensive [17]. One way to mitigate these disadvantages is using OW as an estimator of fish age [16], although the calibration of age is required, such as the traditional otolith reading as a “true age”, whereas age from OW is considered as an “estimate age”. Alternatively, the modal progression of OW or otolith-weight distribution of the captive population and wild-caught fish or the reference sample of known-age fish reared in a hatchery can be used [3,17,18,22]. Unless an exact value for the fish age is available, any method based on OW requires an age estimate from otoliths (or another structure). We used the OW of *S. lalandi* to estimate size at the age under reared conditions; as a result, we knew the exact age (in days) of each fish studied, and calibration was not necessary. As previously stated [26], the ageing of *S. lalandi* requires a precise estimate of the first zone in order to validate estimates for all age classes. Similarly, and as previously described [37], the identification of the first ring of the otoliths of wild *S. lalandi* arriving in Chile was difficult, and a clear and accurate estimate of the first zone was not possible.

Growth parameters for *S. lalandi* were estimated in wild populations from Australia, New Zealand, Japan and South Africa using vertebrae (17th), length frequency distribution, tags and otoliths, and scale reads (Table 2). All those studies estimated the parameters of the von Bertalanffy growth function; our analysis is the only one that includes four different models for this species. The best model, according the Akaike information criterion, was the seasonalized von Bertalanffy growth function. As stated [40,43,44,45], the seasonalized von Bertalanffy growth function has been widely used to fit the growth of fish populations from temperate regions. Therefore, accounting for seasonality in growth is essential for understanding the ecology and management of fish.

Our results suggest that previously, the growth parameters for this fish species were overestimated when seasonality was not considered.

## 5. Conclusions

Our results show that the growth of *Seriola lalandi*, calculated from the growth rate in the weight of the otoliths from a captive population of an exact known age, was best explained by the seasonalized von Bertalanffy growth function. This methodology avoided the need for a calibration factor estimated from the analysis of growth annuli; that is, the “true” age, as suggested [17], was not necessary, because in our study we knew the “real age”.

## Figures and Tables

**Figure 1 animals-12-01640-f001:**
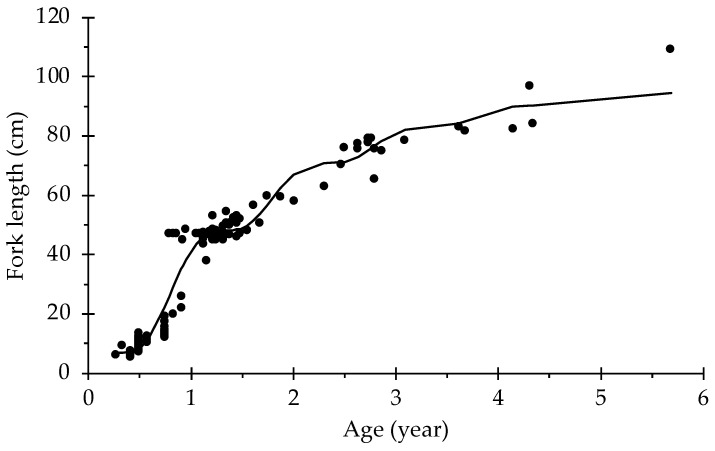
Seasonalized von Bertalanffy growth function for *Seriola lalandi*, estimated by the captive and natural populations.

**Table 1 animals-12-01640-t001:** Values for the growth parameter from the four fitted models. vBGF, von Bertalanffy growth function; GZ, Gompertz model; SvBGF, seasonalized vBGF and LG, logistic model; SE, standard error; AIC, Akaike information criterion.

Model	Parameter	Value	SE	*p*-Value	AIC
vBGF	*L* _∞_	94.18	2.799	<0.0001	602.08
	*K*	0.71	0.044	<0.0001	
	*t* _0_	0.34	0.012	<0.0001	
GZ	*L* _∞_	81.99	1.691	<0.0001	630.11
	G	1.71	0.074	<0.0001	
	to	0.93	0.021	<0.0001	
SvBGF	*L* _∞_	98.58	2.981	<0.0001	533.87
	*K*	0.59	0.04	<0.0001	
	*t* _0_	0.07	0.039	0.086	
	C	0.97	0.131	<0.0001	
	*t* _s_	0.84	0.016	<0.0001	
LG	*L* _∞_	77.76	1.589	<0.0001	673.42
	G	2.86	0.123	<0.0001	
	to	1.13	0.023	<0.0001	

**Table 2 animals-12-01640-t002:** Growth parameters for *Seriola lalandi* from different localities. L_∞,_ asymptotic fork length; *K*, growth coefficient (1/time); *t*_0_, theoretical age when length is zero. Age = estimated age range; Loc. = locality; Ref = reference.

*L* _∞_	*K*	*t* _0_	Age	Method	Loc	Ref	Notes
110.8	0.309	−0.588	1–8	Vertebra read (17th)	Japan	[33]	
125.2	0.189	−0.735	1–6	Length frequency	Australia	[30]	
*	*	*	1–9	Otoliths and scales	Australia	[30]	
*	*	*	1–9	Vertebra read	Australia	[30]	
141.9	0.13		1–11	Tag	New Zealand	[37]	
140.6	0.096	−1.339	4–23	Otolith read	New Zealand	[37]	
106.4	0.173	−2.75	1–8	Otolith read	South Africa	[34]	
116.4	0.247	−0.708	1–29	Otolith read	Northland Gulf/NZ	[45]	Male
131.06	0.172	−0.156	1–29	Otolith read	Northland Gulf/NZ	[45]	Female
120.3	0.184	−1.316	1–29	Otolith read	Bay of Plenty/NZ	[45]	Male
129.6	0.173	−1.074	1–29	Otolith read	Bay of Plenty/NZ	[45]	Female
98.58	0.59	0.07	0.8–5.7	Otolith weight	Northern Chile	This study	

* = According to [46].

## Data Availability

Data are available in Appendix A: Data from captive fishes.

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
