# Peer review of "Otolith Weight as an Estimator of the Age of Seriola lalandi Valenciennes, 1833 (Carangidae), in the Southeastern Pacific"

_animals, 2022, doi:10.3390/ani12131640_

Round 1
Reviewer 1 Report
Present MS of Ndjamba et al. covers the sensitive topic of age estimation in fishes using weight of saggitae. Age estimation in fishes is an essential procedure for management of fisheries resources and stocks assessment, giving information needed for the understanding of populations dynamics, such as recruitments, reproductive and growth dynamics, and mortality. Improving methodologies of age evaluation is very important for a better management and conservation of fish stocks, especially for those over exploited by fisheries and most sensitive to habitats depletion. Develop new technics to avoid the human error improving the speed of age estimation, such as the developing of useful models that leverage the relationships between sagittae weights and age in different species, is fundamental to improve knowledge on populations age structure and consequently the entire fisheries science research field.
However, I have major concerns about MS which should be resolved into being considered for publication. All the MS chapters are poorly organized and written in a confusing way, I strongly suggest a general rewriting process. The aim and the tackled research questions are unclear, and the results chapter is chaotic, especially in the first part, making MS for readers extremely difficult to get the main point of the study. The Introduction needs a general rewriting. It is not well organized, lacking a general Otoliths description for readers less familiar with this research field. The Materials and Methods chapter is also confused, especially in the formula’s descriptions of the different growth functions. The discussion is too concise, with redundant information still present in Introduction, less comments on Results and less comparison with relevant literature. The Conclusion chapter is to be rewritten, giving more importance to future prospective opened by present paper. Moreover, I strongly suggest a professional English proofreading. There are many grammatical errors, too long and unclear phrases, and missing punctuation.
SPECIFIC COMMENTS:
INTRODUCTION:
Lines 44: Please rephrase with: “An accurate estimation age structure and growth patterns of fish population”
Lines 52 – 56: This part is too long and missing the punctuation. Please rewrite it in a clearer way.
Line 59: I strongly suggest adding general information on Otoliths morphology, physiological role, morpho functionality, writing also about all their scientific application fields (such as zoology, ecology, fisheries science, paleontology, and many others). It is also important to provide the relevant literature, to make the text understandable also by readers less familiar with this research fields. Here some literature suggestions:
- Schulz‐Mirbach, Tanja, et al. "Enigmatic ear stones: what we know about the functional role and evolution of fish otoliths." Biological Reviews 94.2 (2019): 457-482.
- D’Iglio, Claudio, et al. "Otolith analyses highlight morpho-functional differences of three species of mullet (Mugilidae) from transitional water." Sustainability1 (2021): 398.
- Mahe, Kelig, et al. "Otolith shape as a valuable tool to evaluate the stock structure of swordfish Xiphias gladius in the Indian Ocean." African Journal of marine science 38.4 (2016): 457-464.
- D’Iglio, Claudio, et al. "Intra-and interspecific variability among congeneric Pagellus otoliths." Scientific reports 11.1 (2021): 1-15.
- Tuset, Víctor M., et al. "Testing otolith morphology for measuring marine fish biodiversity." Marine and Freshwater Research 67.7 (2016): 1037-1048.
- Kéver, Loïc, et al. "Hearing capacities and otolith size in two ophidiiform species (Ophidion rochei and Carapus acus)." Journal of Experimental Biology 217.14 (2014): 2517-2525.
- Chaine, J. & Duvergier, J. Recherches sur les otolithes des poisons. Etude déscriptive et comparative de la sagitta des téléostéens. Act Soc Linn 86, 1–254 (1934).
- Nolf, D. Otolithi Piscium. Handbook of Paleoichthyology, Vol. 10. (1985).
- Volpedo, A. & DianaEcheverría, D. Ecomorphological patterns of the sagitta in fish on the continental shelf off Argentine. Res. 60, 551–560 (2003).
Lines 60 – 62: The sentence is unclear. The part “the otolith growth continues but in thickness” is not connected with the rest. I don’t understand its meaning.
Lines 72-74: The sentence is unclear, please rewrite
Lines 79-80: The sentence is unclear, please rewrite
Line 91: In this part authors should add the general information on the studied species, shifting here the section in Line 99-103. Moreover, I strongly suggest adding also more information and data about the exploitation of S. lalandi, writing in full its scientific name with the authorities (Seriola lalandi, Valenciennes, 1833) the first time, while S. lalandi after.
Lines 105-108: I strongly suggest improving and clarify the aim of this study and the prospective and implication for future development.
MATHERIALS AND METHODS:
Line 113: Why was the Total Length not used? Please explain.
Lines 116-117: Why did not you do a comparison between farmed and wild specimens to assess if in one of the two categories there were a side dimorphism between left and right otoliths?
Lines 124-125: This line should be written in a clearer way, please reword it.
Lines 134-135/138-140/143-144/148-149: I strongly suggest rewriting more clearly the formulas descriptions. Example: where Lt is fork length at age t, L¥ is the asymptotic fork length……
RESULTS:
Lines 155-162: This Results section is confused, please rewrite it. Moreover, samples collection has been dealt with previously (in Materials and Methods). I suggest deleting them.
Author Response
Reviewer 1
Present MS of Ndjamba et al. covers the sensitive topic of age estimation in fishes using weight of saggitae. Age estimation in fishes is an essential procedure for management of fisheries resources and stocks assessment, giving information needed for the understanding of populations dynamics, such as recruitments, reproductive and growth dynamics, and mortality. Improving methodologies of age evaluation is very important for a better management and conservation of fish stocks, especially for those over exploited by fisheries and most sensitive to habitats depletion. Develop new technics to avoid the human error improving the speed of age estimation, such as the developing of useful models that leverage the relationships between sagittae weights and age in different species, is fundamental to improve knowledge on populations age structure and consequently the entire fisheries science research field.
However, I have major concerns about MS which should be resolved into being considered for publication. All the MS chapters are poorly organized and written in a confusing way, I strongly suggest a general rewriting process. The aim and the tackled research questions are unclear, and the results chapter is chaotic, especially in the first part, making MS for readers extremely difficult to get the main point of the study. The Introduction needs a general rewriting. It is not well organized, lacking a general Otoliths description for readers less familiar with this research field. The Materials and Methods chapter is also confused, especially in the formula’s descriptions of the different growth functions. The discussion is too concise, with redundant information still present in Introduction, less comments on Results and less comparison with relevant literature. The Conclusion chapter is to be rewritten, giving more importance to future prospective opened by present paper. Moreover, I strongly suggest a professional English proofreading. There are many grammatical errors, too long and unclear phrases, and missing punctuation.
Manuscript was checked for proper English by MDPI English editing service
SPECIFIC COMMENTS:
INTRODUCTION:
Lines 44: Please rephrase with: “An accurate estimation age structure and growth patterns of fish population”
- Done
Lines 52 – 56: This part is too long and missing the punctuation. Please rewrite it in a clearer way.
- R. We checked this part and change punctuation.
Line 59: I strongly suggest adding general information on Otoliths morphology, physiological role, morpho functionality, writing also about all their scientific application fields (such as zoology, ecology, fisheries science, paleontology, and many others). It is also important to provide the relevant literature, to make the text understandable also by readers less familiar with this research fields. Here some literature suggestions:
- Schulz‐Mirbach, Tanja, et al. "Enigmatic ear stones: what we know about the functional role and evolution of fish otoliths." Biological Reviews 94.2 (2019): 457-482.
- D’Iglio, Claudio, et al. "Otolith analyses highlight morpho-functional differences of three species of mullet (Mugilidae) from transitional water." Sustainability1 (2021): 398.
- Mahe, Kelig, et al. "Otolith shape as a valuable tool to evaluate the stock structure of swordfish Xiphias gladius in the Indian Ocean." African Journal of marine science 38.4 (2016): 457-464.
- D’Iglio, Claudio, et al. "Intra-and interspecific variability among congeneric Pagellus otoliths." Scientific reports 11.1 (2021): 1-15.
- Tuset, Víctor M., et al. "Testing otolith morphology for measuring marine fish biodiversity." Marine and Freshwater Research 67.7 (2016): 1037-1048.
- Kéver, Loïc, et al. "Hearing capacities and otolith size in two ophidiiform species (Ophidion rochei and Carapus acus)." Journal of Experimental Biology 217.14 (2014): 2517-2525.
- Chaine, J. & Duvergier, J. Recherches sur les otolithes des poisons. Etude déscriptive et comparative de la sagitta des téléostéens. Act Soc Linn 86, 1–254 (1934).
- Nolf, D. Otolithi Piscium. Handbook of Paleoichthyology, Vol. 10. (1985).
- Volpedo, A. & DianaEcheverría, D. Ecomorphological patterns of the sagitta in fish on the continental shelf off Argentine. Res. 60, 551–560 (2003).
- A paragraph including additional information, as requested by the reviewer was added, focusing in otoliths as a tool for population and ecological studies, more than aspects related to physiology, which is out the scope of this contribution.
Lines 60 – 62: The sentence is unclear. The part “the otolith growth continues but in thickness” is not connected with the rest. I don’t understand its meaning.
- Sentence was changed.
Lines 72-74: The sentence is unclear, please rewrite
- Sentence changed.
Lines 79-80: The sentence is unclear, please rewrite
- Sentence changed.
Line 91: In this part authors should add the general information on the studied species, shifting here the section in Line 99-103. Moreover, I strongly suggest adding also more information and data about the exploitation of S. lalandi, writing in full its scientific name with the authorities (Seriola lalandi, Valenciennes, 1833) the first time, while S. lalandi after.
- As stated in the MS, knowledge of this species in Chile is scarce and basic biological information is not available.
Lines 105-108: I strongly suggest improving and clarify the aim of this study and the prospective and implication for future development.
MATHERIALS AND METHODS:
Line 113: Why was the Total Length not used? Please explain.
- All the four references related to age and growth of S.lalandi used Fork Length instead Total length.
Lines 116-117: Why did not you do a comparison between farmed and wild specimens to assess if in one of the two categories there were a side dimorphism between left and right otoliths?
R: No side dimorphism was evaluated, but left and right OW, for farmed and wild fish do not differ significantly. Dimorphism or otolith shape analyses are out the scope of this MS.
Lines 124-125: This line should be written in a clearer way, please reword it.
R: Sentence was re-written.
Lines 134-135/138-140/143-144/148-149: I strongly suggest rewriting more clearly the formulas descriptions. Example: where Lt is fork length at age t, L¥ is the asymptotic fork length……
R: Done
RESULTS:
Lines 155-162: This Results section is confused, please rewrite it. Moreover, samples collection has been dealt with previously (in Materials and Methods). I suggest deleting them.
R results were re-writen
Reviewer 2 Report
Dear Authors, please find specific comments below:
- I don’t know what the difference between Simple Summary and Abstract is but there is a quite repetition between these two parts. In addition, please provide more specific results from your study in the section Abstract.
- Line 35 – Please, write species name in italic.
- Line 36 – You say known age, however, only ages of the fish from aquaculture facilities were known. So, these models were calculated only for this group of fish or…?
- Line 38 – 98.58 cm (. instead of ,)
- Line 114-115 – What do you mean by posteriorly otoliths?
- Line 116-117 – Why only for 132 pairs? You excluded the broken one or...?
- Line 119-120 – Revise the sentence.
- Line 125 – You state ‘Due the absence of difference…’ – is this your result or data from some previous studies?
- Line 159 – 37.9 (. instead of ,)
- Results and Discussion – There is a quite big difference in length ranges of the fish from both groups. This can especially be a problem when you calculate otolith growth rate for fish from the aquaculture and then estimate the age for wild population using this parameter in the equation (different length ranges, different environments, different nutrition, different fish metabolic and growth rate, different otolith growth rate…). On the other hand, you emphasize that there is no need of a calibration, and I can’t understand why. In addition, Linf in your study is below the maximum length of fish that you have collected. You don’t discuss this result, but you state that previous estimated growth parameters for this species are overestimated (lines 224-225) – are you sure that they all overestimated growth of Seriola species? There is a repetition of some data from Introduction in the Discussion; please, revise this part. Moreover, lines 212-214, please add a reference for this note.
Author Response
Reviewer 2
Dear Authors, please find specific comments below:
I don’t know what the difference between Simple Summary and Abstract is but there is a quite repetition between these two parts. In addition, please provide more specific results from your study in the section Abstract.
- Main results are included in the abstract. Number and range size of fish from aquaculture facilities as well as for wild population were included
Line 35 – Please, write species name in italic.
- Done
Line 36 – You say known age, however, only ages of the fish from aquaculture facilities were known. So, these models were calculated only for this group of fish or…?
- We know the exact age of fish from aquaculture facilities, from this data we calculated the otolith growth rate and using this value we estimate the age of fish from the wild population, as stated in the Material and Methods.
Line 38 – 98.58 cm (. instead of ,)
- Changed
Line 114-115 – What do you mean by posteriorly otoliths?
- Sentence changed.
Line 116-117 – Why only for 132 pairs? You excluded the broken one or...?
- Broken otoliths were discarded. A short sentenced was added.
Line 119-120 – Revise the sentence
- Sentence was changed.
Line 125 – You state ‘Due the absence of difference…’ – is this your result or data from some previous studies?
- We feel that the position of references generate the doubt. Position of references was changed and now is clear that the data come from references 34 and 37.
Line 159 – 37.9 (. instead of ,)
- Changed
Results and Discussion – There is a quite big difference in length ranges of the fish from both groups. This can especially be a problem when you calculate otolith growth rate for fish from the aquaculture and then estimate the age for wild population using this parameter in the equation (different length ranges, different environments, different nutrition, different fish metabolic and growth rate, different otolith growth rate…).
- We are confident that there is a difference in length ranges from both groups, but the smaller fish from wild population was 37.9 cm. Small fishes are not found in the northern Chilean coast and the origin of the migrant is not known. In a similar way, fish larger than 25.7 cm were not available at aquaculture facilities. Our approach is the only way to get estimates of age - growth for this species.
On the other hand, you emphasize that there is no need of a calibration, and I can’t understand why.
- When OW is used to estimate age of the fish, without the knowledge of the precise age of the fish, a calibration is needed, but we know the exact age of fish from aquaculture facilities.
In addition, Linf in your study is below the maximum length of fish that you have collected. You don’t discuss this result, but you state that previous estimated growth parameters for this species are overestimated (lines 224-225) – are you sure that they all overestimated growth of Seriola species?
- Linf is a parameter calculated from a series of data. Is just an estimator of the maximum size a fish can reach and not a true or exact value of the maximum (asymptotic) size. Please note that for the same data set, four different values for Linf are estimated from four different models (Table 1).The same apply for the other growth parameters. We stated that “Our results suggest that previously, the growth parameters for this fish species were overestimated when seasonality was not considered “.
There is a repetition of some data from Introduction in the Discussion; please, revise this part.
Moreover, lines 212-214, please add a reference for this note.
- A reference was added.
Round 2
Reviewer 1 Report
Dear Authors, I have read your answers to my comments, clarifying the critical points of the manuscript. I have nothing more to ask and, in my opinion, the paper can be accepted by the journal
Author Response
Dear reviewer
Thank you very much for your comments and suggestions that helped to significantly improve our contribution
Reviewer 2 Report
Dear Authors,
Thank you for your responses and the revision of the manuscript.
I'm still not convinced that it is correct to use two groups of fish, that is, to calculate otolith growth rate for fish from the aquaculture and then estimate the age for wild population using this parameter in the equation. It is clear to me when you say that you do not have larger specimens in the aquaculture and that you cannot catch smaller individuals in the wild, but that is not the reason for choosing this methodology. In addition, regarding your note that growth parameters for this species were overestimated when seasonality was not considered, since your samples originate from January to April, I don’t see how you included seasonality in your age estimates (using seasonalized VBGF doesn’t mean that). And yes, you used four different aging methods but which methods were used by the authors listed in Table 2?
Author Response
I'm still not convinced that it is correct to use two groups of fish, that is, to calculate otolith growth rate for fish from the aquaculture and then estimate the age for wild population using this parameter in the equation. It is clear to me when you say that you do not have larger specimens in the aquaculture and that you cannot catch smaller individuals in the wild, but that is not the reason for choosing this methodology
R: In the Discussion section we stated:
"As previously stated [26], the ageing of Seriola lalandi requires a precise estimate of the first zone in order to validate estimates for all age classes. Similary, and as previously described [37], the identification of the first ring of otoliths of wild S. lalandi arriving in Chile was difficult, and a clear and accurate estimate of the first zone was not possible".
This is a good reason the choose the described metodology. No way to estimate age from otolith rings with other methodology.
In addition, regarding your note that growth parameters for this species were overestimated when seasonality was not considered, since your samples originate from January to April, I don’t see how you included seasonality in your age estimates (using seasonalized VBGF doesn’t mean that).
R: The seasonalized vBGF is just a model; that is, an approach to estimate growth parameters. This model DO NOT require seasonal samples. Please note that we are working with models. The best non linear fit for our data is represented by the seasonalized vBGF.
And yes, you used four different aging methods but which methods were used by the authors listed in Table 2?
R: Please note that in the legend of Table 2 we indicated the methods followed by the cited authors.